# Gaussian Process Volatility Model

**Yue Wu**
Cambridge University
wu5@post.harvard.edu

**José Miguel Hernández Lobato**
Cambridge University
jmh233@cam.ac.uk

**Zoubin Ghahramani**
Cambridge University
zoubin@eng.cam.ac.uk

## Abstract

The prediction of time-changing variances is an important task in the modeling of financial data. Standard econometric models are often limited as they assume rigid functional relationships for the evolution of the variance. Moreover, functional parameters are usually learned by maximum likelihood, which can lead to over-fitting. To address these problems we introduce GP-Vol, a novel non-parametric model for time-changing variances based on Gaussian Processes. This new model can capture highly flexible functional relationships for the variances. Furthermore, we introduce a new online algorithm for fast inference in GP-Vol. This method is much faster than current offline inference procedures and it avoids overfitting problems by following a fully Bayesian approach. Experiments with financial data show that GP-Vol performs significantly better than current standard alternatives.

## 1 Introduction

Time series of financial returns often exhibit heteroscedasticity, that is the standard deviation or volatility of the returns is time-dependent. In particular, large returns (either positive or negative) are often followed by returns that are also large in size. The result is that financial time series frequently display periods of low and high volatility. This phenomenon is known as volatility clustering [1]. Several univariate models have been proposed in the literature for capturing this property. The best known and most popular is the Generalised Autoregressive Conditional Heteroscedasticity model (GARCH) [2]. An alternative to GARCH are stochastic volatility models [3]. However, there is no evidence that SV models have better predictive performance than GARCH [4, 5, 6].

GARCH has further inspired a host of variants and extensions. A review of many of these models can be found in [7]. Most of these GARCH variants attempt to address one or both limitations of GARCH: a) the assumption of a linear dependency between current and past volatilities, and b) the assumption that positive and negative returns have symmetric effects on volatility. Asymmetric effects are often observed, as large negative returns often send measures of volatility soaring, while this effect is smaller for large positive returns [8, 9]. Finally, there are also extensions that use additional data besides daily closing prices to improve volatility predictions [10].

Most solutions proposed in these variants of GARCH involve: a) introducing nonlinear functional relationships for the evolution of volatility, and b) adding asymmetric effects in these functional relationships. However, the GARCH variants do not fundamentally address the problem that the specific functional relationship of the volatility is unknown. In addition, these variants can have a high number of parameters, which may lead to overfitting when using maximum likelihood learning.

More recently, volatility modeling has received attention within the machine learning community, with the development of copula processes [11] and heteroscedastic Gaussian processes [12]. These

models leverage the flexibility of Gaussian Processes [13] to model the unknown relationship between the variances. However, these models do not address the asymmetric effects of positive and negative returns on volatility.

We introduce a new non-parametric volatility model, called the Gaussian Process Volatility Model (GP-Vol). This new model is more flexible, as it is not limited by a fixed functional form. Instead, a non-parametric prior distribution is placed on possible functions, and the functional relationship is learned from the data. This allows GP-Vol to explicitly capture the asymmetric effects of positive and negative returns on volatility. Our new volatility model is evaluated in a series of experiments with real financial returns, and compared against popular econometric models, namely, GARCH, EGARCH [14] and GJR-GARCH [15]. In these experiments, GP-Vol produces the best overall predictions. In addition to this, we show that the functional relationship learned by GP-Vol often exhibits the nonlinear and asymmetric features that previous models attempt to capture.

The second main contribution of the paper is the development of an online algorithm for learning GP-Vol. GP-Vol is an instance of a Gaussian Process State Space Model (GP-SSM). Previous work on GP-SSMs [16, 17, 18] has mainly focused on developing approximation methods for filtering and smoothing the hidden states in GP-SSM, without jointly learning the GP transition dynamics. Only very recently have Frigola et al. [19] addressed the problem of learning both the hidden states and the transition dynamics by using Particle Gibbs with Ancestor Sampling (PGAS) [20]. In this paper, we introduce a new online algorithm for performing inference on GP-SSMs. Our algorithm has similar predictive performance as PGAS on financial data, but is much faster.

## 2 Review of GARCH and GARCH variants

The standard variance model for financial data is GARCH. GARCH assumes a Gaussian observation model and a linear transition function for the variance: the time-varying variance $\sigma_t^2$ is linearly dependent on $p$ previous variance values and $q$ previous squared time series values, that is,

$$x_t \sim \mathcal{N}(0, \sigma_t^2), \qquad \text{and} \qquad \sigma_t^2 = \alpha_0 + \sum_{j=1}^q \alpha_j x_{t-j}^2 + \sum_{i=1}^p \beta_i \sigma_{t-i}^2, \qquad (1)$$

where $x_t$ are the values of the return time series being modeled. This model is flexible and can produce a variety of clustering behaviors of high and low volatility periods for different settings of $\alpha_1, \ldots, \alpha_q$ and $\beta_1, \ldots, \beta_p$. However, it has several limitations. First, only linear relationships between $\sigma_{t-p:t-1}^2$ and $\sigma_t^2$ are allowed. Second, past positive and negative returns have the same effect on $\sigma_t^2$ due to the quadratic term $x_{t-j}^2$. However, it is often observed that large negative returns lead to larger rises in volatility than large positive returns [8, 9].

A more flexible and often cited GARCH extension is Exponential GARCH (EGARCH) [14]. The equation for $\sigma_t^2$ is now:

$$\log(\sigma_t^2) = \alpha_0 + \sum_{j=1}^q \alpha_j g(x_{t-j}) + \sum_{i=1}^p \beta_i \log(\sigma_{t-i}^2), \quad \text{where} \quad g(x_t) = \theta x_t + \lambda |x_t|. \quad (2)$$

Asymmetry in the effects of positive and negative returns is introduced through the function $g(x_t)$. If the coefficient $\theta$ is negative, negative returns will increase volatility, while the opposite will happen if $\theta$ is positive. Another GARCH extension that models asymmetric effects is GJR-GARCH [15]:

$$\sigma_t^2 = \alpha_0 + \sum_{j=1}^q \alpha_j x_{t-j}^2 + \sum_{i=1}^p \beta_i \sigma_{t-i}^2 + \sum_{k=1}^r \gamma_k x_{t-k}^2 I_{t-k}, \qquad (3)$$

where $I_{t-k} = 0$ if $x_{t-k} \geq 0$ and $I_{t-k} = 1$ otherwise. The asymmetric effect is now captured by $I_{t-k}$, which is nonzero if $x_{t-k} < 0$.

## 3 Gaussian process state space models

GARCH, EGARCH and GJR-GARCH can be all represented as General State-Space or Hidden Markov models (HMM) [21, 22], with the unobserved dynamic variances being the hidden states. Transition functions for the hidden states are fixed and assumed to be linear in these models. The linear assumption limits the flexibility of these models.

More generally, a non-parametric approach can be taken where a Gaussian Process (GP) prior is placed on the transition function, so that its functional form can be learned from data. This Gaussian Process state space model (GP-SSM) is a generalization of HMM. GP-SSM and HMM differ in two main ways. First, in HMM the transition function has a fixed functional form, while in GP-SSM

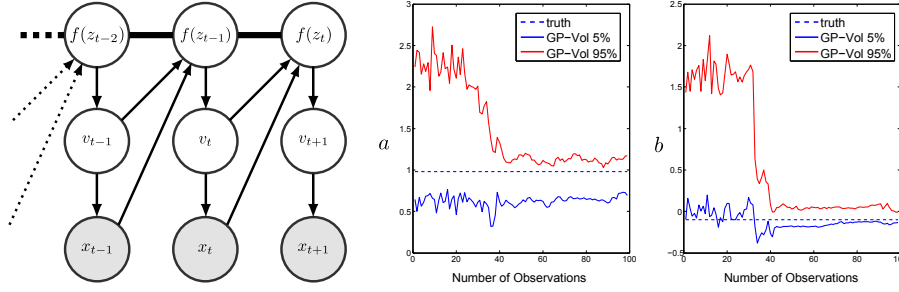

Figure 1: Left, graphical model for GP-Vol. The transitions of the hidden states $v_t$ is represented by the unknown function $f$. $f$ takes as inputs the previous state $v_{t-1}$ and previous observation $x_{t-1}$. Middle, 90% posterior interval for $a$. Right, 90% posterior interval for $b$.

it is represented by a GP. Second, in GP-SSM the states do not have Markovian structure once the transition function is marginalized out.

The flexibility of GP-SSMs comes at a cost: inference in GP-SSMs is computationally challenging. Because of this, most of the previous work on GP-SSMs [16, 17, 18] has focused on filtering and smoothing the hidden states in GP-SSM, without jointly learning the GP dynamics. Note that in [18], the authors learn the dynamics, but using a separate dataset in which both input and target values for the GP model are observed. A few papers considered learning both the GP dynamics and the hidden states for special cases of GP-SSMs. For example, [23] applied EM to obtain maximum likelihood estimates for parametric systems that can be represented by GPs. A general method has been recently proposed for joint inference on the hidden states and the GP dynamics using Particle Gibbs with Ancestor Sampling (PGAS) [20, 19]. However, PGAS is a batch MCMC inference method that is computationally very expensive.

## 4  Gaussian process volatility model

Our new Gaussian Process Volatility Model (GP-Vol) is an instance of GP-SSM:
$$x_t \sim \mathcal{N}(0, \sigma_t^2), \qquad v_t := \log(\sigma_t^2) = f(v_{t-1}, x_{t-1}) + \epsilon_t, \qquad \epsilon_t \sim \mathcal{N}(0, \sigma_n^2). \qquad (4)$$
Note that we model the logarithm of the variance, which has real support. Equation (4) defines a GP-SMM. We place a GP prior on the transition function $f$. Let $z_t = (v_t, x_t)$. Then $f \sim \mathcal{GP}(m, k)$ where $m(z_t)$ and $k(z_t, z_t')$ are the GP mean and covariance functions, respectively. The mean function can encode prior knowledge of the system dynamics. The covariance function gives the prior covariance between function values: $k(z_t, z_t') = \mathrm{Cov}(f(z_t), f(z_t'))$. Intuitively if $z_t$ and $z_t'$ are close to each other, the covariances between the corresponding function values should be large: $f(z_t)$ and $f(z_t')$ should be highly correlated.

The graphical model for GP-Vol is given in Figure 1. The explicit dependence of transition function values on the previous return $x_{t-1}$ enables GP-Vol to model the asymmetric effects of positive and negative returns on the variance evolution. GP-Vol can be extended to depend on $p$ previous log variances and $q$ past returns like in GARCH($p$,$q$). In this case, the transition would be of the form $v_t = f(v_{t-1}, v_{t-2}, ..., v_{t-p}, x_{t-1}, x_{t-2}, ..., x_{t-q}) + \epsilon_t$.

## 5  Bayesian inference in GP-Vol

In the standard GP regression setting, the inputs and targets are fully observed and $f$ can be learned using exact Bayesian inference [13]. However, this is not the case in GP-Vol, where the unknown $\{v_t\}$ form part of the inputs and all the targets. Let $\boldsymbol{\theta}$ denote the model hyper-parameters and let $\mathbf{f} = [f(v_1), \ldots, f(v_T)]$. Directly learning the joint posterior of the unknown variables $\mathbf{f}$, $v_{1:T}$ and $\boldsymbol{\theta}$ is a challenging task. Fortunately, the posterior $p(v_t|\boldsymbol{\theta}, x_{1:t})$, where $\mathbf{f}$ has been marginalized out, can be approximated with particles [24]. We first describe a standard sequential Monte Carlo (SMC) particle filter to learn this posterior.

Let $\{v_{1:t-1}^i\}_{i=1}^N$ be particles representing chains of states up to $t-1$ with corresponding normalized weights $W_{t-1}^i$. The posterior $p(v_{1:t-1}|\boldsymbol{\theta}, x_{1:t-1})$ is then approximated by
$$\hat{p}(v_{1:t-1}|\boldsymbol{\theta}, x_{1:t-1}) = \sum_{i=1}^N W_{t-1}^i \delta_{v_{1:t-1}^i}(v_{1:t-1}). \qquad (5)$$

The corresponding posterior for $v_{1:t}$ can be approximated by propagating these particles forward. For this, we propose new states from the GP-Vol transition model and then we importance-weight them according to the GP-Vol observation model. Specifically, we resample particles $v_{1:t-1}^j$ from (5) according to their weights $W_{t-1}^j$, and propagate the samples forward. Then, for each of the particles propagated forward, we propose $v_t^j$ from $p(v_t|\boldsymbol{\theta}, v_{1:t-1}^j, x_{1:t-1})$, which is the GP predictive distribution. The proposed particles are then importance-weighted according to the observation model, that is, $W_t^j \propto p(x_t|\boldsymbol{\theta}, v_t^j) = \mathcal{N}(x_t|0, \exp\{v_t^j\})$.

The above setup assumes that $\boldsymbol{\theta}$ is known. To learn these hyper-parameters, we can also encode them in particles and filter them together with the hidden states. However, since $\boldsymbol{\theta}$ is constant across time, naively filtering such particles without regeneration will fail due to particle impoverishment, where a few or even one particle receives all the weight. To solve this problem, the Regularized Auxiliary Particle Filter (RAPF) regenerates parameter particles by performing kernel smoothing operations [25]. This introduces artificial dynamics and estimation bias. Nevertheless, RAPF has been shown to produce state-of-the-art inference in multivariate parametric financial models [6].

RAPF was designed for HMMs, but GP-Vol is non-Markovian once $\mathbf{f}$ is marginalized out. Therefore, we design a new version of RAPF for non-Markovian systems and refer to it as the Regularized Auxiliary Particle Chain Filter (RAPCF), see Algorithm 1. There are two main parts in RAPCF. First, there is the Auxiliary Particle Filter (APF) part in lines 5, 6 and 7 of the pseudocode [26]. This part selects particles associated with high expected likelihood, as given by the new expected state in (7) and the corresponding resampling weight in (8). This bias towards particles with high expected likelihood is eliminated when the final importance weights are computed in (9). The most promising particles are propagated forward in lines 8 and 9. The main difference between RAPF and RAPCF is in the effect that previous states $v_{1:t-1}^i$ have in the propagation of particles. In RAPCF all the previous states determine the probabilities of the particles being propagated, as the model is non-Markovian, while in RAPF these probabilities are only determined by the last state $v_{t-1}^i$. The second part of RAPCF avoids particle impoverishment in $\boldsymbol{\theta}$. For this, new particles are generated in line 10 by sampling from a Gaussian kernel. The over-dispersion introduced by these artificial dynamics is eliminated in (6) by shrinking the particles towards their empirical average. We fix the shrinking parameter $\lambda$ to be $0.95$. In practice, we found little difference in predictions when we varied $\lambda$ from $0.99$ to $0.95$.

RAPCF has limitations similar to those of RAPF. First, it introduces bias as sampling from the kernel adds artificial dynamics. Second, RAPCF only filters forward and does not smooth backward. Consequently, there will be impoverishment in distant ancestors $v_{t-L}$, since these states are not regenerated. When this occurs, GP-Vol will consider the collapsed ancestor states as inputs with little uncertainty and the predictive variance near these inputs will be underestimated. These issues can be addressed by adopting a batch MCMC approach. In particular, Particle Markov Chain Monte Carlo (PMCMC) procedures [24] established a framework for learning the states and the parameters in general state space models. Additionally, [20] developed a PMCMC algorithm called Particle Gibbs with ancestor sampling (PGAS) for learning non-Markovian state space models. PGAS was applied by [19] to learn GP-SSMs. These batch MCMC methods are computationally much more expensive than RAPCF. Furthermore, our experiments show that in the GP-Vol model, RAPCF and PGAS have similar empirical performance, while RAPCF is orders of magnitude faster than PGAS. This indicates that the aforementioned issues have limited impact in practice.

## 6 Experiments

We performed three sets of experiments. First, we tested on synthetic data whether we can jointly learn the hidden states and transition dynamics in GP-Vol using RAPCF. Second, we compared the performance of GP-Vol against standard econometric models GARCH, EGARCH and GJR-GARCH on fifty real financial time series. Finally, we compared RAPCF with the batch MCMC method PGAS in terms of accuracy and execution time. The code for RAPCF in GP-Vol is publicly available at `http://jmhl.org`.

### 6.1 Experiments with synthetic data

We generated ten synthetic datasets of length $T = 100$ according to (4). The transition function $f$ is sampled from a GP prior specified with a linear mean function and a squared exponential covariance

**Algorithm 1** RAPCF

1: **Input:** data $x_{1:T}$, number of particles $N$, shrinkage parameter $0 < \lambda < 1$, prior $p(\boldsymbol{\theta})$.
2: Sample $N$ parameter particles from the prior: $\{\boldsymbol{\theta}_0^i\}_{i=1,...,N} \sim p(\boldsymbol{\theta})$.
3: Set initial importance weights, $W_0^i = 1/N$.
4: **for** $t = 1$ **to** $T$ **do**
5:    Shrink parameter particles towards their empirical mean $\bar{\boldsymbol{\theta}}_{t-1} = \sum_{i=1}^N W_{t-1}^i \boldsymbol{\theta}_{t-1}^i$ by setting
$$\widetilde{\boldsymbol{\theta}}_t^i = \lambda \boldsymbol{\theta}_{t-1}^i + (1 - \lambda)\bar{\boldsymbol{\theta}}_{t-1} \,. \tag{6}$$
6:    Compute the new expected states:
$$\boldsymbol{\mu}_t^i = \mathbb{E}(v_t | \widetilde{\boldsymbol{\theta}}_t^i, v_{1:t-1}^i, x_{1:t-1}) \,. \tag{7}$$
7:    Compute importance weights proportional to the likelihood of the new expected states:
$$g_t^i \propto W_{t-1}^i p(x_t | \boldsymbol{\mu}_t^i, \widetilde{\boldsymbol{\theta}}_t^i) \,. \tag{8}$$
8:    Resample $N$ auxiliary indices $\{j\}$ according to weights $\{g_t^i\}$.
9:    Propagate the corresponding chains of hidden states forward, that is, $\{v_{1:t-1}^j\}_{j \in J}$.
10:   Add jitter: $\boldsymbol{\theta}_t^j \sim \mathcal{N}(\widetilde{\boldsymbol{\theta}}_t^j, (1 - \lambda^2)\mathbf{V}_{t-1})$, where $\mathbf{V}_{t-1}$ is the empirical covariance of $\boldsymbol{\theta}_{t-1}$.
11:   Propose new states $v_t^j \sim p(v_t | \boldsymbol{\theta}_t^j, v_{1:t-1}^j, x_{1:t-1})$.
12:   Compute importance weights adjusting for the modified proposal:
$$W_t^j \propto p(x_t | v_t^j, \boldsymbol{\theta}_t^j) / p(x_t | \boldsymbol{\mu}_t^j, \widetilde{\boldsymbol{\theta}}_t^j) \,, \tag{9}$$
13: **end for**
14: **Output:** particles for chains of states $v_{1:T}^j$, particles for parameters $\boldsymbol{\theta}_t^j$ and particle weights $W_t^j$.

---

function. The linear mean function is $\mathbb{E}(v_t) = m(v_{t-1}, x_{t-1}) = av_{t-1} + bx_{t-1}$. The squared exponential covariance function is $k(y, z) = \gamma \exp(-0.5|y - z|^2/l^2)$ where $l$ is the length-scale parameter and $\gamma$ is the amplitude parameter.

We used RAPCF to learn the hidden states $v_{1:T}$ and the hyper-parameters $\boldsymbol{\theta} = (a, b, \sigma_n, \gamma, l)$ using non-informative diffuse priors for $\boldsymbol{\theta}$. In these experiments, RAPCF successfully recovered the state and the hyper-parameter values. For the sake of brevity, we only include two typical plots of the 90% posterior intervals for hyper-parameters $a$ and $b$ in the middle and right of Figures 1. The intervals are estimated from the filtered particles for $a$ and $b$ at each time step $t$. In both plots, the posterior intervals eventually concentrate around the true parameter values, shown as dotted blue lines.

## 6.2 Experiments with real data

We compared the predictive performances of GP-Vol, GARCH, EGARCH and GJR-GARCH on real financial datasets. We used GARCH(1,1), EGARCH(1,1) and GJR-GARCH(1,1,1) models since these variants have the least number of parameters and are consequently less affected by overfitting problems. We considered fifty datasets, consisting of thirty daily Equity and twenty daily foreign exchange (FX) time series. For the Equity series, we used daily closing prices. For FX, which operate 24h a day, with no official daily closing prices, we cross-checked different pricing sources and took the consensus price up to 4 decimal places at 10am New York, which is the time with most market liquidity. Each of the resulting time series contains a total of $T = 780$ observations from January 2008 to January 2011. The price data $p_{1:T}$ was pre-processed to eliminate prices corresponding to times when markets were closed or not liquid. After this, prices were converted into logarithmic returns, $x_t = \log(p_t/p_{t-1})$. Finally, the resulting returns were standardized to have zero mean and unit standard deviation.

During the experiments, each method receives an initial time series of length 100. The different models are trained on that data and then a one-step forward prediction is made. The performance of each model is measured in terms of the predictive log-likelihood on the first return out of the training set. Then the training set is augmented with the new observation and the training and prediction steps are repeated. The whole process is repeated sequentially until no further data is received.

GARCH, EGARCH and GJR-GARCH were implemented using numerical optimization routines provided by Kevin Sheppard [1]. A relatively long initial time series of length 100 was needed to to train these models. Using shorter initial data resulted in wild jumps in the maximum likelihood

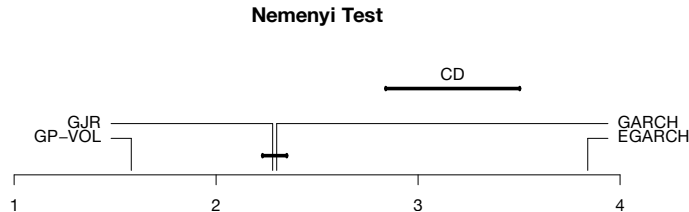

**Nemenyi Test**

Figure 2: Comparison between GP-Vol, GARCH, EGARCH and GJR-GARCH via a Nemenyi test. The figure shows the average rank across datasets of each method (horizontal axis). The methods whose average ranks differ more than a critical distance (segment labeled CD) show significant differences in performance at this confidence level. When the performances of two methods are statistically different, their corresponding average ranks appear disconnected in the figure.

estimates of the model parameters. These large fluctuations produced very poor one-step forward predictions. By contrast, GP-Vol is less susceptible to overfitting since it approximates the posterior distribution using RAPCF instead of finding point estimates of the model parameters. We placed broad non-informative priors on $\boldsymbol{\theta} = (a, b, \sigma_n, \gamma, l)$ and used $N = 200$ particles and shrinkage parameter $\lambda = .95$ in RAPCF.

| Dataset | GARCH | EGARCH | GJR | GP-Vol |
|---|---|---|---|---|
| AUDUSD | −1.303 | −1.514 | −1.305 | **−1.297** |
| BRLUSD | −1.203 | −1.227 | −1.201 | **−1.180** |
| CADUSD | −1.402 | −1.409 | −1.402 | **−1.386** |
| CHFUSD | −1.375 | −1.404 | −1.404 | **−1.359** |
| CZKUSD | −1.422 | −1.473 | **−1.422** | −1.456 |
| EURUSD | −1.418 | −2.120 | −1.426 | **−1.403** |
| GBPUSD | **−1.382** | −3.511 | −1.386 | −1.385 |
| IDRUSD | −1.223 | −1.244 | −1.209 | **−1.039** |
| JPYUSD | −1.350 | −2.704 | −1.355 | **−1.347** |
| KRWUSD | −1.189 | −1.168 | −1.209 | **−1.154** |
| MXNUSD | −1.220 | −3.438 | −1.278 | **−1.167** |
| MYRUSD | −1.394 | −1.412 | −1.395 | **−1.392** |
| NOKUSD | −1.416 | −1.567 | −1.419 | **−1.416** |
| NZDUSD | **−1.369** | −3.036 | −1.379 | −1.389 |
| PLNUSD | −1.395 | −1.385 | **−1.382** | −1.393 |
| SEKUSD | −1.403 | −3.705 | **−1.402** | −1.407 |
| SGDUSD | **−1.382** | −2.844 | −1.398 | −1.393 |
| TRYUSD | **−1.224** | −1.461 | −1.238 | −1.236 |
| TWDUSD | −1.384 | −1.377 | −1.388 | **−1.294** |
| ZARUSD | −1.318 | −1.344 | **−1.301** | −1.304 |

Table 1: FX series.

| Dataset | GARCH | EGARCH | GJR | GP-Vol |
|---|---|---|---|---|
| A | −1.304 | −1.449 | **−1.281** | −1.282 |
| AA | −1.228 | −1.280 | −1.230 | **−1.218** |
| AAPL | −1.234 | −1.358 | −1.219 | **−1.212** |
| ABC | −1.341 | −1.976 | −1.344 | **−1.337** |
| ABT | **−1.295** | −1.527 | −1.3003 | −1.302 |
| ACE | −1.084 | −2.025 | −1.106 | **−1.073** |
| ADBE | −1.335 | −1.501 | −1.386 | **−1.302** |
| ADI | −1.373 | −1.759 | **−1.352** | −1.356 |
| ADM | −1.228 | −1.884 | **−1.223** | −1.223 |
| ADP | −1.229 | −1.720 | **−1.205** | −1.211 |
| ADSK | −1.345 | −1.604 | −1.340 | **−1.316** |
| AEE | −1.292 | −1.282 | −1.263 | **−1.166** |
| AEP | −1.151 | −1.177 | −1.146 | **−1.142** |
| AES | −1.237 | −1.319 | −1.234 | **−1.197** |
| AET | −1.285 | −1.302 | −1.269 | **−1.246** |

Table 2: Equity series 1-15.

| Dataset | GARCH | EGARCH | GJR | GP-Vol |
|---|---|---|---|---|
| AFL | −1.057 | −1.126 | −1.061 | **−0.997** |
| AGN | −1.270 | −1.338 | **−1.261** | −1.274 |
| AIG | −1.151 | −1.256 | −1.195 | **−1.069** |
| AIV | **−1.111** | −1.147 | −1.1285 | −1.133 |
| AIZ | −1.423 | −1.816 | −1.469 | **−1.362** |
| AKAM | −1.230 | −1.312 | **−1.229** | −1.246 |
| AKS | −1.030 | −1.034 | −1.052 | **−1.015** |
| ALL | −1.339 | −3.108 | **−1.316** | −1.327 |
| ALTR | −1.286 | −1.443 | **−1.277** | −1.282 |
| AMAT | −1.319 | −1.465 | −1.332 | **−1.310** |
| AMD | −1.342 | −1.348 | −1.332 | **−1.243** |
| AMGN | −1.191 | −1.542 | **−1.1772** | −1.189 |
| AMP | −1.386 | −1.444 | −1.365 | **−1.317** |
| AMT | **−1.206** | −1.820 | −1.3658 | −1.210 |
| AMZN | **−1.206** | −1.567 | −1.3537 | −1.342 |

Table 3: Equity series 16-30.

We show the average predictive log-likelihood of GP-Vol, GARCH, EGARCH and GJR-GARCH in tables 1, 2 and 3 for the FX series, the first 15 Equity series and the last 15 Equity series, respectively. The results of the best performing method in each dataset have been highlighted in bold. These tables show that GP-Vol obtains the highest predictive log-likelihood in 29 of the 50 analyzed datasets. We perform a statistical test to determine whether differences among GP-Vol, GARCH, EGARCH and GJR-GARCH are significant. These methods are compared against each other using the multiple comparison approach described by [27]. In this comparison framework, all the methods are ranked according to their performance on different tasks. Statistical tests are then applied to determine whether the differences among the average ranks of the methods are significant. In our case, each of the 50 datasets analyzed represents a different task. A Friedman rank sum test rejects the hypothesis that all methods have equivalent performance at $\alpha = 0.05$ with $p$-value less than $10^{-15}$. Pairwise comparisons between all the methods with a Nemenyi test at a 95% confidence level are summarized in Figure 2. The Nemenyi test shows that GP-Vol is significantly better than the other methods.

The other main advantage of GP-Vol over existing models is that it can learn the functional relationship $f$ between the new log variance $v_t$ and the previous log variance $v_{t-1}$ and previous return $x_{t-1}$. We plot a typical log variance surface in the left of Figure 3. This surface is generated by plotting the mean predicted outputs $v_t$ against a grid of inputs for $v_{t-1}$ and $x_{t-1}$. For this, we use the functional dynamics learned with RAPCF on the AUDUSD time series. AUDUSD stands for the amount of US dollars that an Australian dollar can buy. The grid of inputs is designed to contain a range of values experienced by AUDUSD from 2008 to 2011, which is the period covered by the data. The surface is colored according to the standard deviation of the posterior predictive distribution for the log variance. Large standard deviations correspond to uncertain predictions, and are redder.

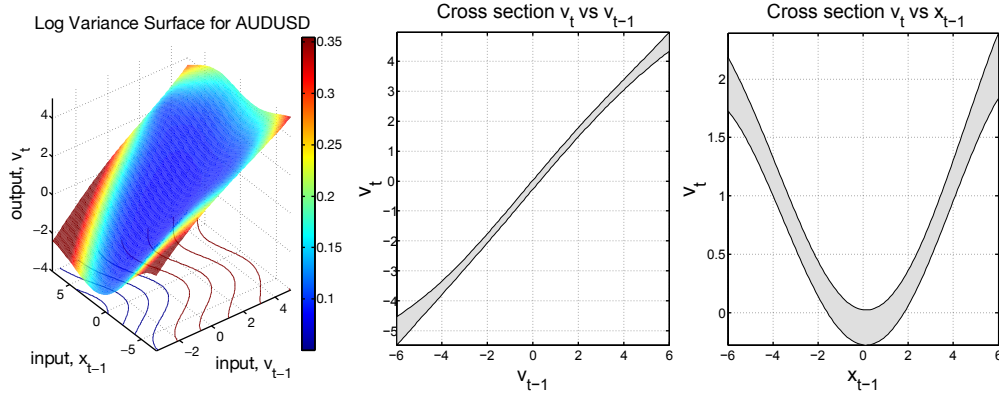

Figure 3: Left, surface generated by plotting the mean predicted outputs $v_t$ against a grid of inputs for $v_{t-1}$ and $x_{t-1}$. Middle, predicted $v_t \pm 2$ s.d. for inputs $(0, x_{t-1})$. Right, predicted $v_t \pm 2$ s.d. for inputs $(0, x_{t-1})$.

The plot in the left of Figure 3 shows several patterns. First, there is an asymmetric effect of positive and negative previous returns $x_{t-1}$. This can be seen in the skewness and lack of symmetry of the contour lines with respect to the $v_{t-1}$ axis. Second, the relationship between $v_{t-1}$ and $v_t$ is slightly non-linear because the distance between consecutive contour lines along the $v_{t-1}$ axis changes as we move across those lines, especially when $x_{t-1}$ is large. In addition, the relationship between $x_{t-1}$ and $v_t$ is nonlinear, but some sort of skewed quadratic function. These two patterns confirm the asymmetric effect and the nonlinear transition function that EGARCH and GJR-GARCH attempt to model. Third, there is a dip in predicted log variance for $v_{t-1} < -2$ and $-1 < x_{t-1} < 2.5$. Intuitively this makes sense, as it corresponds to a calm market environment with low volatility. However, as $x_{t-1}$ becomes more extreme the market becomes more turbulent and $v_t$ increases.

To further understand the transition function $f$ we study cross sections of the log variance surface. First, $v_t$ is predicted for a grid of $v_{t-1}$ and $x_{t-1} = 0$ in the middle plot of Figure 3. Next, $v_t$ is predicted for various $x_{t-1}$ and $v_{t-1} = 0$ in the right plot of Figure 3. The confidence bands in the figures correspond to the mean prediction $\pm 2$ standard deviations. These cross sections confirm the nonlinearity of the transition function and the asymmetric effect of positive and negative returns on the log variance. The transition function is slightly non-linear as a function of $v_{t-1}$ as the band in the middle plot of Figure 3 passes through $(-2, -2)$ and $(0, 0)$, but not $(2, 2)$. Surprisingly, we observe in the right plot of Figure 3 that large positive $x_{t-1}$ produces larger $v_t$ when $v_{t-1} = 0$ since the band is slightly higher at $x_{t-1} = 6$ than at $x_{t-1} = -6$. However, globally, the highest predicted $v_t$ occurs when $v_{t-1} > 5$ and $x_{t-1} < -5$, as shown in the surface plot.

## 6.3 Comparison between RAPCF and PGAS

We now analyze the potential shortcomings of RAPCF that were discussed in Section 5. For this, we compare RAPCF against PGAS on the twenty FX time series from the previous section in terms of predictive log-likelihood and execution times. The RAPCF setup is the same as in Section 6.2. For PGAS, which is a batch method, the algorithm is run on initial training data $x_{1:L}$, with $L = 100$, and a one-step forward prediction is made. The predictive log-likelihood is evaluated on the next observation out of the training set. Then the training set is augmented with the new observation and the batch training and prediction steps are repeated. The process is repeated sequentially until no further data is received. For these experiments we used shorter time series with $T = 120$ since PGAS is computationally very expensive. Note that we cannot simply learn the GP-SSM dynamics on a small set of training data and then predict on a large test dataset, as it was done in [19]. These authors were able to predict forward as they were using synthetic data with known "hidden" states.

We analyze different settings of RAPCF and PGAS. In RAPCF we use $N = 200$ particles since that number was used to compare against GARCH, EGARCH and GJR-GARCH in the previous section. PGAS has two parameters: a) $N$, the number of particles and b) $M$, the number of iterations. Three combinations of these settings were used. The resulting average predictive log-likelihoods for RAPCF and PGAS are shown in Table 4. On each dataset, the results of the best performing method

have been highlighted in bold. The average rank of each method across the analyzed datasets is shown in Table 5. From these tables, there is no evidence that PGAS outperforms RAPCF on these financial datasets, since there is no clear predictive edge of any PGAS setting over RAPCF.

| | RAPCF | PGAS.1 | PGAS.2 | PGAS.3 |
|---|---|---|---|---|
| | $N = 200$ | $N = 10$ | $N = 25$ | $N = 10$ |
| **Dataset** | | $M = 100$ | $M = 100$ | $M = 200$ |
| AUDUSD | $-1.1205$ | $\mathbf{-1.0571}$ | $-1.0699$ | $-1.0936$ |
| BRLUSD | $-1.0102$ | $-1.0043$ | $-0.9959$ | $\mathbf{-0.9759}$ |
| CADUSD | $-1.4174$ | $-1.4778$ | $-1.4514$ | $\mathbf{-1.4077}$ |
| CHFUSD | $\mathbf{-1.8431}$ | $-1.8536$ | $-1.8453$ | $-1.8478$ |
| CZKUSD | $-1.2263$ | $-1.2357$ | $-1.2424$ | $\mathbf{-1.2093}$ |
| EURUSD | $-1.3837$ | $-1.4586$ | $\mathbf{-1.3717}$ | $-1.4064$ |
| GBPUSD | $-1.1863$ | $-1.2106$ | $-1.1790$ | $\mathbf{-1.1729}$ |
| IDRUSD | $-0.5446$ | $\mathbf{-0.5220}$ | $-0.5388$ | $-0.5463$ |
| JPYUSD | $-2.0766$ | $\mathbf{-1.9286}$ | $-2.1585$ | $-2.1658$ |
| KRWUSD | $\mathbf{-1.0566}$ | $-1.1212$ | $-1.2032$ | $-1.2066$ |
| MXNUSD | $-0.2417$ | $-0.2731$ | $\mathbf{-0.2271}$ | $-0.2538$ |
| MYRUSD | $\mathbf{-1.4615}$ | $-1.5464$ | $-1.4745$ | $-1.4724$ |
| NOKUSD | $-1.3095$ | $-1.3443$ | $\mathbf{-1.3048}$ | $-1.3169$ |
| NZDUSD | $-1.2254$ | $\mathbf{-1.2101}$ | $-1.2366$ | $-1.2373$ |
| PLNUSD | $-0.8972$ | $\mathbf{-0.8704}$ | $-0.8708$ | $\mathbf{-0.8704}$ |
| SEKUSD | $-1.0085$ | $\mathbf{-1.0085}$ | $-1.0505$ | $-1.0360$ |
| SGDUSD | $\mathbf{-1.6229}$ | $-1.9141$ | $-1.7566$ | $-1.7837$ |
| TRYUSD | $\mathbf{-1.8336}$ | $-1.8509$ | $-1.8352$ | $-1.8553$ |
| TWDUSD | $\mathbf{-1.7093}$ | $-1.7178$ | $-1.8315$ | $-1.7257$ |
| ZARUSD | $\mathbf{-1.3236}$ | $-1.3326$ | $-1.3440$ | $-1.3286$ |

Table 4: Results for RAPCF vs. PGAS.

| Method | Configuration | Rank |
|---|---|---|
| RAPCF | $N = 200$ | **2.025** |
| PGAS.1 | $N = 10, M = 100$ | 2.750 |
| PGAS.2 | $N = 25, M = 100$ | 2.550 |
| PGAS.3 | $N = 10, M = 200$ | 2.675 |

Table 5: Average ranks.

| Method | Configuration | Avg. Time |
|---|---|---|
| RAPCF | $N = 200$ | **6** |
| PGAS.1 | $N = 10, M = 100$ | 732 |
| PGAS.2 | $N = 25, M = 100$ | 1832 |
| PGAS.3 | $N = 10, M = 200$ | 1465 |

Table 6: Avg. running time.

As mentioned above, there is little difference between the predictive accuracies of RAPCF and PGAS. However, PGAS is computationally much more expensive. We show average execution times in minutes for RAPCF and PGAS in Table 6. Note that RAPCF is up to two orders of magnitude faster than PGAS. The cost of this latter method could be reduced by using fewer particles $N$ or fewer iterations $M$, but this would also reduce its predictive accuracy. Even after doing so, PGAS would still be more costly than RAPCF. RAPCF is also competitive with GARCH, EGARCH and GJR, whose average training times are in this case 2.6, 3.5 and 3.1 minutes, respectively. A naive implementation of RAPCF has cost $O(NT^4)$, since at each time step $t$ there is a $O(T^3)$ cost from the inversion of the GP covariance matrix. On the other hand, the cost of applying PGAS naively is $O(NMT^5)$, since for each batch of data $x_{1:t}$ there is a $O(NMT^4)$ cost. These costs can be reduced to be $O(NT^3)$ and $O(NMT^4)$ for RAPCF and PGAS respectively by doing rank one updates of the inverse of the GP covariance matrix at each time step. The costs can be further reduced by a factor of $T^2$ by using sparse GPs [28].

## 7 Summary and discussion

We have introduced a novel Gaussian Process Volatility model (GP-Vol) for time-varying variances in financial time series. GP-Vol is an instance of a Gaussian Process State-Space model (GP-SSM) which is highly flexible and can model nonlinear functional relationships and asymmetric effects of positive and negative returns on time-varying variances. In addition, we have presented an online inference method based on particle filtering for GP-Vol called the Regularized Auxiliary Particle Chain Filter (RAPCF). RAPCF is up to two orders of magnitude faster than existing batch Particle Gibbs methods. Results for GP-Vol on 50 financial time series show significant improvements in predictive performance over existing models such as GARCH, EGARCH and GJR-GARCH. Finally, the nonlinear transition functions learned by GP-Vol can be easily analyzed to understand the effect of past volatility and past returns on future volatility.

For future work, GP-Vol can be extended to learn the functional relationship between a financial instrument's volatility, its price and other market factors, such as interest rates. The functional relationship thus learned can be useful in the pricing of volatility derivatives on the instrument. Additionally, the computational efficiency of RAPCF makes it an attractive choice for inference in other GP-SSMs different from GP-Vol. For example, RAPCF could be more generally applied to learn the hidden states and the dynamics in complex control systems.

## Footnotes

[1] http:///www.kevinsheppard.com/wiki/UCSD_GARCH/

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
