[Reviews · NeurIPS 2014]

Submitted by Assigned_Reviewer_25

The paper introduces a GP-Vol model to flexibly capture the time-dependent changes in variance, and develops a new online algorithm for fully Bayesian inference under the model.

The paper is clearly written, the developed inference method seems technically sound, and the presented results look promising. My opinion on the model itself, using a non-parametric approach such as using the GP prior on the transition function (as in the paper), seems, though, a bit an obvious way of extending the prior work developed in the finance area. So, I wouldn't put too high grade on the paper in terms of its originality. However, overall the paper is nicely presented and convincing.

Here are a few questions to the authors:

(1) the savings in computation time compared the other methods looks very impressive, and results all look very promising. Have authors also tested higher order dependency on variances and returns, i.e., higher p and q than just 1? How is it different from the order-1 dependency in terms of computation time and algorithmic efficiency? can authors learn the optimal values for p and q from the data, maybe by computing test likelihood?

(2) authors mentions bias introduced from the sampling from artificial dynamics in kernel. Have the authors quantify the bias? It would be more convincing to know when/how RAPCF fails, rather than just saying "the issues have limited impacts in practice".

(3) I am not sure how authors can choose the shrinkage parameter, lambda, in practice.

Summary: In this paper, the evolution of the time-varying variance is modeled by GP and the associated fully Bayesian inference method is presented. Overall, it is a well-written and technically sound paper.

Submitted by Assigned_Reviewer_29

This paper proposes a new nonparametric volatility model.

The contribution of the paper relies on the application of Gaussian process prior to stochastic volatility models. The inference approach is based on SMC and is a straightforward application of the Liu and West's APF algorithm.

The literature review lacks of some important references and the empirical comparison is not complete. My major concerns are the following:
\begin{enumerate}
\item The authors proposed a new stochastic volatility (SV) model, but in the introduction and in the paper there is no reference (e.g., Taylor (1987), Jacquier, Polson, Rossi (1994, 2004) to this class of models.
\item A comparison is proposed between GP-Vol and GARCH models, but the increased flexibility of a SV over the GARCH family has already been proved in the literature by many parts. See e.g. Fridman and Harris (1998). Thus I would strongly suggest the inclusion of a comparison between a standard SV model and the proposed GP-Vol model.
\item The author proposed a new nonparametric model, but no references is give to the recent advances in Bayesian nonparametric which make use of flexible prior processes such as Dirichlet processes (e.g., see Bassetti, Casarin, Leisen (2014), Griffin (2011), Griffin and Steel (2011)).
\end{enumerate}

References:
\begin{itemize}
\item Jacquier, E., Polson, N., Rossi, P., 1994. Bayesian analysis of stochastic volatility models (with discussion). Journal of Business and Economic Statistics 12 (4), 371-417.

\item Jacquier, E., Polson, N., Rossi, P., 1994. Bayesian analysis of stochastic volatility models with fat-tails and correlated errors. Journal of Econometrics 122, 185-212.

\item Fridman, M., Harris, L., 1998. A maximum likelihood approach for non-Gaussian stochastic volatility models.
Journal of Business and Economics Statistics 16 (3), 284-291.

\item Bassetti, F., Casarin, R., Leisen, F. (2014), Pitman-Yor Process Prior for Bayesian Inference, Journal of Econometrics, 180, 49-72.

\item J. E. Griffin. Inference in infinite superpositions of non-Gaussian Ornstein-Uhlenbeck processes
using Bayesian nonparametic methods. Journal of Financial Econometrics, 1:1-31, 2011.

\item J. E. Griffin and M. F. J. Steel. Stick-breaking autoregressive processes. Journal of Econometrics, 162:383-396, 2011.
\end{itemize}
Summary: I would expect the authors improve the presentation of their contribution and provide a comparison with a standard SV model. I would suggest to accept the paper after revision.

Submitted by Assigned_Reviewer_41

This paper provides an alternative based on Gaussian processes (GP) to standard GARCH-related methods for modeling the time-varying volatility of financial time series. The benefit of the GP approach is greater flexibility.

This submission is technically competent, but not of broad interest or great novelty. The mathematics and the Bayesian algorithm are specialized for this problem.

The experiments look rather unfair. The synthetic datasets are well-specified for GPs and mis-specified for GARCH variants, so it is not surprising that GPs do well. For real data "We used GARCH(1,1), EGARCH(1,1) and GJR-GARCH(1,1,1) models since these variants have the least number of parameters and are consequently less affected by overfitting" It seems there was no attempt to regularize GARCH variants, or to choose the best level of complexity for them. Even so, the authors method performs best on only 58% of datasets.

The author of this review is an experienced professor at a research university and "data science" director in industry. My advice to the author(s) is to apply their strong technical skills to a broader problem, and/or a less theoretical application.

COMMENTS ON THE AUTHORS' RESPONSE

The response seems valid technically to me, so I have upgraded my numerical evaluation of the paper. I still believe that the paper is not of broad interest in machine learning. This reviewer has served on the PhD committee of students in econometrics. If this paper is of interest to that community, it should be published there.

The authors say that higher-order GARCH models do no better than almost-trivial GARCH models. The GP method here does better, but not dramatically so. As a practitioner in applied finance, I say that all these models are better as mathematics than as descriptions of reality. The research area needs to go in a different direction, back from mathematical sophistication towards insight into facts.
Summary: Too narrow a problem.
Author Feedback
Author rebuttal: We thank the reviewers for their interesting comments and careful analysis. We are also pleased to receive some strong positive feedback.

Reviewer_41:

“Submission is not of broad interest”:

The prediction of time-changing variances is of high interest in machine learning and econometrics. The 2003 Nobel prize in economics was awarded to Robert F. Engle for his work on the ARCH process, a predecessor of the GARCH model. The GARCH paper (Bollerslev, 1986) has about 16000 citations, with 1500 only in the last year. Moreover, a large number of papers on the modeling of time-changing variances have also been recently published in machine learning venues. A few recent examples are

Variational Heteroscedastic Gaussian Process Regression. In ICML 2011.
Generalised Wishart processes. In UAI, 2011.
Copula processes. In NIPS, 2010.
Efficient Variational Inference for Gaussian Process Regression Networks. In AISTATS 2013.
Gaussian process regression networks. In ICML 2012,
Dynamic Covariance Models for Multivariate Financial Time Series. In ICML, 2013.

Furthermore, our algorithm RAPCF has direct applications to other Gaussian Process State Space Models (GP-SSMs) distinct from GP-Vol. GP-SSMs are used in many areas of science, engineering and economics to model time series and dynamical systems, see for example (Frigola et al. 2013). Note that RAPCF is much cheaper computationally than the state-of-the-art Particle Gibbs with ancestor sampling (PGAS) used by Frigola et al. 2013 and in our experiments RAPCF has similar predictive performance to PGAS.

Therefore, we believe our contribution is of broad interest and wide applicability.

“Submission is not of great novelty”:

Our approach is up to our knowledge the first application of Gaussian Process State Models to describe the time-changing variance in financial time series. Furthermore, the proposed method RAPCF is, again to our knowledge, the first successful application of online particle filters to GP-SSMs. Therefore, we believe our contribution is significantly novel.

“The mathematics and the Bayesian algorithm are specialized for this problem”

We disagree. As described above, our algorithm RAPCF has direct applications to other GP-SSMs different from GP-Vol. GP-SSMs are used in many areas of science, engineering and economics to model time series and dynamical systems.

“synthetic datasets are well-specified for GPs and mis-specified for GARCH variants”

The reviewer probably misunderstood our experiments with synthetic data. We do not compare with GARCH variants in these experiments. These experiments are included to validate our inference algorithm RAPCF. They allow us to verify that the posterior approximation generated by RAPCF concentrates around the true value of the model parameters. For this reason, these experiments include data sampled from the assumed GP-Vol model. We do not compare with GARCH variants in these experiments as it would not be fair since they assume a less general generative model. All the other experiments in which we compare GP-Vol with the GARCH variants use real-world data.

“It seems there was no attempt to regularize GARCH variants, or to choose the best level of complexity for them”

We did study higher order dependencies. In practice, GARCH(p,q) with p and q larger than 1 underperform GARCH(1,1) on a large number of datasets. Therefore for a clean comparison of GP-Vol, we focused on GP-Vol(1,1) vs GARCH(1,1).

We are unaware of any study showing how to regularize GARCH and that the regularized version of GARCH significantly outperforms the standard GARCH(1,1) version. We would appreciate if the reviewer could point us to that study in his final response.

“the authors method performs best on only 58% of datasets”

The figure 58% is misleading because we are comparing 4 different methods. If all the methods performed the same, we would expect our method to win only 25% of the times. The 58% value that we obtain is much higher than that. Furthermore, our multiple comparison test which is highly conservative shows that GP-Vol is statistically the best method at the 5% level.

Reviewer_25:

“Have the authors tested higher order dependency, i.e., higher p and q than just 1?”:

See our one but last response to Reviewer_41.

"bias introduced by the artificial dynamics?”

Liu and West (1999) show that the degradation due to bias is negligible for series of size 50. We performed further extensive experiments. The bias problem is negligible for series with about 1000 observations, but there is some degradation for longer series. In practice, the cost of GPs limits these models to series with about 1000 observations so the bias problem is not significant for the applications that we are considering.

“How do authors choose the shrinkage parameter \lambda?”:

One could use cross-validation ideas to pick the best shrinkage parameter. In practice, we found little difference in predictions when we varied the shrinkage parameter from 0.99 to 0.95.

Reviewer_29:

“No references to stochastic volatility models”:

We will add references to the relevant papers in the final draft.

“Comparison with SV models”:

Despite the flexibility of SV models, there is no evidence that SV models have better predictive performance than GARCH models (Poon and Granger (2005), Kim et al (2005), Wu et al (2013)). We will include a comparison of SV vs GP-Vol in the final version of the paper.

Poon, S. & Granger, C. (2005). Practical issues in forecasting volatility, Financial Analysts Journal, 61, 45–56.

K., Sangjoon, N. Shephard, and S. Chib. "Stochastic volatility: likelihood inference and comparison with ARCH models." The Review of Economic Studies 65.3 (1998): 361-393.